# A study protocol to characterise pathophysiological and molecular markers of rheumatic heart disease and degenerative aortic stenosis using multiparametric cardiovascular imaging and multiomics techniques

**Daniel W. Mutithu**[1,2,3], **Olukayode O. Aremu**[1,2,3], **Dipolelo Mokaila**[1,2,3], **Tasnim Bana**[1,2,3], **Mary Familusi**[2,4], **Laura Taylor**[5], **Lorna J. Martin**[5], **Laura J. Heathfield**[5], **Jennifer A. Kirwan**[6,7], **Lubbe Wiesner**[8], **Henry A. Adeola**[9], **Evelyn N. Lumngwena**[1,2,3,10], **Rodgers Manganyi**[11], **Sebastian Skatulla**[4], **Richard Naidoo**[12], **Ntobeko A. B. Ntusi**[1,2,3,13,14] *

**1** Department of Medicine, Cape Heart Institute, University of Cape Town, Cape Town, South Africa, **2** Division of Cardiology, Department of Medicine, University of Cape Town and Groote Schuur Hospital, Cape Town, South Africa, **3** Extramural Unit on Intersection of Noncommunicable Diseases and Infectious Diseases, South African Medical Research Council, Cape Town, South Africa, **4** Department of Civil Engineering, University of Cape Town, Cape Town, South Africa, **5** Division of Forensic Medicine and Toxicology, Department of Pathology, University of Cape Town, Cape Town, South Africa, **6** Metabolomics Platform, Berlin Institute of Health at Charité – Universitätsmedizin Berlin, Berlin, Germany, **7** Max-Delbrück-Center (MDC) for Molecular Medicine, Helmholtz Association, Berlin, Germany, **8** Division of Clinical Pharmacology, Department of Medicine, University of Cape Town, Cape Town, South Africa, **9** Division of Dermatology, Department of Medicine, University of Cape Town, Cape Town, South Africa, **10** School of Clinical Medicine, University of the Witwatersrand, Johannesburg, South Africa, **11** Chris Barnard Division of Cardiothoracic Surgery, University of Cape Town and Groote Schuur Hospital, Cape Town, South Africa, **12** Division of Anatomical Pathology, Department of Pathology, University of Cape Town and National Health Laboratory Service, Cape Town, South Africa, **13** Cape Universities Body Imaging Centre, University of Cape Town, Cape Town, South Africa, **14** Wellcome Centre for Infectious Disease Research, University of Cape Town, Cape Town, South Africa

* ntobeko.ntusi@uct.ac.za

**Data Availability Statement:** No datasets were generated or analysed during the current study. All

## Abstract

### Introduction

Rheumatic heart disease (RHD), degenerative aortic stenosis (AS), and congenital valve diseases are prevalent in sub-Saharan Africa. Many knowledge gaps remain in understanding disease mechanisms, stratifying phenotypes, and prognostication. Therefore, we aimed to characterise patients through clinical profiling, imaging, histology, and molecular biomarkers to improve our understanding of the pathophysiology, diagnosis, and prognosis of RHD and AS.

relevant data from this study will be made available upon study completion.

**Funding:** This research is funded through a Blue Skies research grant from the National Research Foundation to SS (Grant Numbers 104839 and 105858) https://www.nrf.ac.za/. NABN gratefully acknowledges funding from the National Research Foundation (https://www.nrf.ac.za/), South African Medical Research Council (https://www.samrc.ac.za) and the Lily and Ernst Hausmann Trust. The funders has no role in study design, data collection and analysis, decision to publish, or preparation of the manuscript.

**Competing interests:** The authors have declared that no competing interests exist.

## Methods

In this cross-sectional, case–controlled study, we plan to recruit RHD and AS patients and compare them to matched controls. Living participants will undergo clinical assessment, echocardiography, CMR and blood sampling for circulatory biomarker analyses. Tissue samples will be obtained from patients undergoing valve replacement, while healthy tissues will be obtained from cadavers. Immunohistology, proteomics, metabolomics, and transcriptome analyses will be used to analyse circulatory- and tissue-specific biomarkers. Univariate and multivariate statistical analyses will be used for hypothesis testing and identification of important biomarkers. In summary, this study aims to delineate the pathophysiology of RHD and degenerative AS using multiparametric CMR imaging. In addition to discover novel biomarkers and explore the pathomechanisms associated with RHD and AS through high-throughput profiling of the tissue and blood proteome and metabolome and provide a proof of concept of the suitability of using cadaveric tissues as controls for cardiovascular disease studies.

## Introduction

Valvular heart disease (VHD) in sub-Saharan Africa (SSA) is mainly secondary to rheumatic heart disease (RHD), degenerative aortic stenosis (AS) and congenital VHD, most commonly involving the bicuspid aortic valve (BAV) [1]. The prevalence of RHD is remarkably high in low- and middle-income countries (LMICs), making it the most common cause of heart failure in young individuals in LMICs [2]. According to the Global Burden of Disease Study, the incidence rates of RHD have remained stable (changes in age-standardised prevalence 0–1% between 1990 and 2019), while the incidence rates of nonrheumatic VHD increased between 1990 and 2019 [3]. Coincidentally, central and southern SSA showed a high age-standardised incidence rate of RHD (29.40/100,000) in 2017 and an increase in nonrheumatic VHD from 244.55/100,000 to 247.26/100,000 between 1990 and 2017 [3].

VHD diagnosis is predicated on a thorough clinical history and detailed physical examination. Diagnosis is confirmed on imaging, primarily with transthoracic echocardiography, but other imaging modalities may be used, including cardiovascular magnetic resonance (CMR), cardiovascular computed tomography (CT), fluoroscopy and X-ray techniques, transoesophageal echocardiography, and invasive angiography. In addition, the diagnosis of VHD may be supported using electrocardiography and circulatory biomarkers [4]. According to current guidelines, 2-D transthoracic echocardiography with Doppler is the first-line imaging modality for characterising RHD and degenerative AS patients [4, 5]. CMR is increasingly used for the study of tissue characteristics and prognosis in patients with poor acoustic windows or where there are discrepant echocardiographic and clinical features [6].

Although the incidence of acute rheumatic fever (ARF) has declined in Europe and North America over the past 4–6 decades, RHD remains one of the most important causes of cardiovascular morbidity and mortality among socially and economically disadvantaged populations worldwide, especially in LMICs that are home to most of the world's population [2]. The Jones criteria, which have been the clinical standard for establishing the diagnosis of ARF since 1944, were last modified by the American Heart Association in 1992 [7, 8]. Because echocardiographic techniques have evolved worldwide during the past 2 decades and because echocardiography has become a cornerstone in worldwide screening programs for evaluating the prevalence of RHD, the limited diagnostic role of echocardiography in the diagnosis of carditis

is no longer appropriate. In 2015, the World Heart Federation (WHF) revised the Jones criteria for the diagnosis of ARF [8]. In 2020, the American College of Cardiology (ACC)/American Heart Association (AHA) also revised their guidelines for the diagnosis and management of patients with VHD [4].

Acute rheumatic fever starts from cross-reactivity of the immune response to pharyngeal infection with group A Streptococcus (GAS) to host proteins in genetically susceptible patients. Infection causes the proliferation of GAS-specific T cells and B cells [9, 10]. The *S. pyogenes* antigen and human proteins (myosin, vimentin and/or tropomyosin) share immunological epitopes that lead to immune-mediated tissue injury through molecular mimicry. Infection with GAS and cross-reactive antibodies and T cells leads to common symptoms associated with ARF (carditis, arthritis, chorea, erythema marginatum and subcutaneous nodules) [8]. In addition, inflammation leads to neovascularization which fuels the inflammatory cascade that causes granulomatous lesions [9, 10]. Several studies have indicated that ARF susceptibility could be due to polymorphisms in genes encoding proteins involved in the immune response, suggesting genetic susceptibility to ARF [11]. It is estimated that at least 60% of ARF patients develop RHD with mitral regurgitation and/or stenosis or aortic regurgitation with some cases of aortic stenosis [10, 11]. The number of ARF episodes, age at initial ARF diagnosis, poor adherence to prophylaxis, and severity of carditis at the earliest ARF episode are risk factors associated with progression to chronic RHD [10, 11]. The valvular pathologies characteristic of chronic RHD (leaflet thickening, chordal thickening, and excessive anterior leaflet motion) often progress asymptomatically until severe damage to the valves occurs [10]. The observed valve pathologies in RHD patients are thought to be due to collagen remodelling and calcification. It has been suggested that after GAS infection, an autoimmune reaction is triggered against collagen type IV, which leads to anti-collagen reactions that affect the balance between collagen deposition and degradation, thus causing fibrosis [10]. Although not fully understood, it is postulated that proinflammatory matrix metalloproteinases (MMPs) modulate calcification through the degradation of elastin, which promotes smooth muscle conversion to osteoblastic phenotypes. In addition, calcification is thought to be promoted by neovascularization, wherein vascular endothelial growth factors are expressed, which have a regulatory effect on bone remodelling [10]. Furthermore, neovascularization contributes to the destabilisation of extracellular matrix mechanics and promotes persistent local inflammation [10, 12]. The pathophysiological mechanisms involved in the progression of RHD, especially translational and metabolic processes, are not fully understood [13]. On the other hand, degenerative AS is caused by inflammatory pathways in the context of advanced age, smoking, dyslipidemia, and hypertension [3, 14]. In addition, NOTCH1, Fibrillin-1 (FBN1), and Filamin (AFLNA) gene mutations have been associated with the development of degenerative calcification and mitral valve prolapse [15, 16]. Degenerative aortic stenosis progression is associated with osteoblast differentiation, myofibroblast activation, and/or high shear forces [3, 17]. Furthermore, perturbations in lipid, lipoprotein, and glycerophospholipid levels have been associated with the progression of calcific aortic stenosis [17–20].

Several studies have investigated the involvement of immunopathological [10], cell signalling [21], genetic [22, 23] and translational [13, 24] processes in the pathogenesis of RHD and degenerative AS. In this study, we intend to employ multiparametric CMR, molecular biology, and multiomic analyses of biomarkers to study RHD and degenerative AS.

Autophagy is a cellular clearing mechanism that maintains cellular homeostasis. Autophagy helps the cell survive harmful stimuli such as a lack of nutrients, hypoxia, inflammation, reactive oxygen species (ROS), endoplasmic reticulum stress, and damaged mitochondria. Autophagy dysregulation has been reported in cancer, neurodegeneration, advanced age, and cardiovascular disease (CVD) [25–27]. Decreased autophagy is associated with aging and leads

to giant cardiac mitochondria, reduced adenosine triphosphate (ATP) production and increased ROS levels [28]. The major molecular mediators of autophagy signalling include unc-51-like kinase 1 (ULK1), autophagy-related gene-phosphoinositide-interacting 1 (Atg-WIPI1), and vacuole protein sorting-Beclin 1 (Vps34-Beclin 1) class phosphoinositol-3-phosphate-kinase (PI3-kinase) complexes, autophagy-related gene 12 (Atg12), and protein 1 light chain 3 (LC3) conjugation systems [29]. There is a paucity of data on the role of autophagy biomarkers in the pathophysiology and progression of RHD and degenerative AS. AS has been reported to be associated with excessive deposition of misaligned collagen, glycoproteins, and osteogenic nodules [28]. The activation of autophagy reduces osteogenic activity, reduces calcification, and suppresses inflammation in calcific AS [28]. Elevated autophagy biomarker levels are associated with increased cell death in heart failure [30].

Proteome characterisation in VHD has been explored using proteomics techniques [24, 31], which have reported the involvement of myofibrogenic and oxidative stress-related pathways in calcific AS [31]. Dysregulated plasma complement C4A, carbonic anhydrase, and vitronectin have been reported as potential biomarkers in RHD and AS [24]. Lumican, vitronectin, collagen VI, and vimentin are some of the structural and cellular adhesion proteins dysregulated in RHD [13].

Metabolomics has rarely been used to study VHD; dysregulation of inflammatory processes, energy metabolism, amino acid metabolism, serotonin metabolism, and calcium metabolism have all been associated with myxomatous MS and MR [32, 33]. In addition, metabolic signatures have shown a strong correlation with clinical parameters of valvular morphology, severity, and markers of cardiac injury [34–37]. Formate and lactate have shown very good performance as diagnostic biomarkers of degenerative MS and MR [33]. Urinary glycine, hippurate, and taurine showed good diagnostic performance in differentiating BAV patients from healthy individuals [38]. Dysregulation of arachidonic acid metabolism pathways post-transcatheter aortic valve replacement (post-TAVR) was associated with worse outcomes and reduced reverse remodelling [37]. Despite the high prevalence of RHD and AS in Africa, understanding of their pathogenesis has been elusive. The molecular mechanisms involved in the pathophysiology of RHD are not well understood [10, 39]. In addition, very few studies have compared the molecular mechanisms underlying RHD and degenerative aortic stenosis despite having different etiologies. To the best of our knowledge, very few studies have compared the patterns of circulatory and tissue-specific biomarkers in RHD and degenerative aortic stenosis. Furthermore, it is challenging to explore molecular biomarkers on cardiac tissues due to the challenges and ethical issues surrounding access to healthy tissues as controls [40]. Very few studies have explored suitable postmortem intervals when considering the use of control cadaveric tissues.

## Materials and methods

### Study objectives

This study is designed to characterise VHD patients with mild and severe RHD and degenerative AS based on clinical profiling, CMR, histological parameters, and molecular biomarkers (Fig 1). This study will test the hypothesis that RHD and degenerative AS exhibit distinct pathophysiological mechanisms and follow different molecular processes, as detected by CMR, autophagy, proteomics, and metabolomics profiles. Furthermore, this study will test the hypothesis that there are different molecular signatures associated with mild and severe VHD, and these signatures can be used for early detection. The null hypothesis will be rejected if the differences in the tested variables between the tested groups are significantly different (p< 0.05).

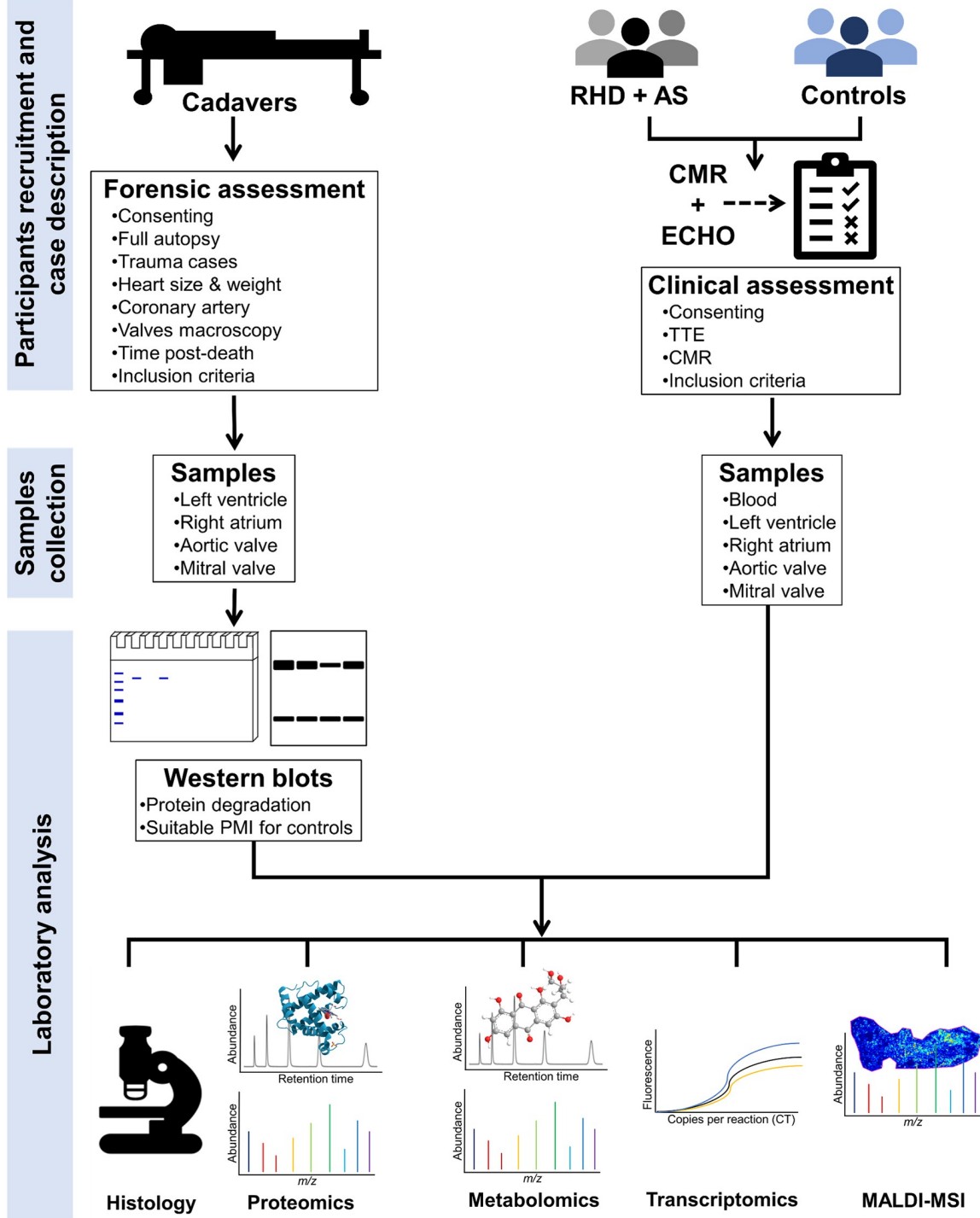

**Fig 1. Schematic summary of the study design.** Patients with rheumatic heart disease (RHD) or degenerative aortic stenosis (AS), healthy controls, and cadaveric tissue donors will be recruited. Cardiovascular phenotypes will be characterised with CMR imaging and 2D-TTE in RHD patients, AS patients and controls. Cadaveric donors will be assessed by forensic pathologists. Samples will be obtained from RHD patients, AS patients, healthy participants, and cadavers. SDS–PAGE and western blot analysis will be used to assess protein degradation postdeath, and suitable cadaver PMIs for use as healthy controls. Histology, proteomics, metabolomics, transcriptomics, and MALDI-MSI will be used for molecular characterisation and colocalisation. (CMR, cardiac magnetic resonance imaging; 2D-TTE, 2D-transthorasic echocardiography; MALDI-MSI, matrix-assisted laser desorption/ionisation imaging mass spectrometry; SDS–PAGE, sodium dodecyl sulfate–polyacrylamide gel electrophoresis; PMIs, postmortem intervals).

The specific objectives of this study are as follows:

- Characterisation of the phenotypes and pathophysiology of RHD and degenerative AS patients using CMR.

- Determination of the immunological processes associated with RHD and degenerative AS using autophagy markers.

- Characterisation of the tissue and circulatory proteome in patients diagnosed with RHD and degenerative AS using proteomics.

- Description of the circulatory and tissue-specific metabolome in RHD and degenerative AS using metabolomics.

- Determination of the molecular profiles associated with mild and severe VHD.

- Description of protein degradation in cardiac tissue from cadaveric samples to determine suitable postmortem intervals in deceased controls for CVD research.

- Description of the colocalisation of metabolic and proteomic biomarkers in VHD histopathology using matrix-assisted laser desorption–ionisation–imaging mass spectrometry (MALDI–MSI).

- To investigate the suitability of using dried plasma spots for sample collection and storage in metabolomics studies.

## Study design and sample size

This cross-sectional, case–controlled study aims to recruit adult participants diagnosed with severe or mild VHD (RHD or degenerative AS) from the Cardiac Clinic at Groote Schuur Hospital (GSH) in Cape Town. GSH is a tertiary/quaternary-level health facility affiliated with the University of Cape Town in the Western Cape Province of South Africa. Recruited patients will be diagnosed by a cardiologist according to the 2015 WHF Revised Jones Criteria and/or the 2020 ACC/AHA guidelines on the diagnosis of VHD [4, 8, 41]. It is challenging to predict *a priori* the required sample size where there are highly divergent test variables from different techniques and platforms (univariate and multivariate) and there have been no prior studies to define the abnormal biomarkers and prevalence in RHD and AS. To establish a suitable sample size, several methods were employed owing to the multidisciplinary nature of the study. PowerEQTL [42] was based on previously published genome-wide association study (GWAS) data, and a sample size of 300 participants was estimated to have a power of 0.80 and a minor allelic frequency (MAF) of 0.11. Genome-wide association studies (GWASs) on rheumatic heart diseases have reported that the top variant of the class III region of the HLA complex has an MAF of 0.15 [43]. We therefore decided to recruit 250 VHD patients (65 RHD severe patients, 65 severe AS patients, 60 mild RHD patients and 60 mild AS patients) and 250 age-, sex-, comorbidity-, and ethnicity-matched healthy participants (200 ambulant adults and 50 deceased adults without cardiovascular disease recruited from the state's Forensic Pathology Services). During recruitment, we will ensure that there is a balance between male and female participants. Eligible patients will be screened and enrolled as per the inclusion and exclusion criteria (Table 1). For the omics studies (genomics, metabolomics, and proteomics studies), participants and control living individuals will undergo CMR at the Cape Universities Body Imaging Centre (CUBIC), University of Cape Town.

**Table 1. Inclusion and exclusion criteria for RHD and AS patients.**

| Inclusion criteria | Exclusion criteria |
| --- | --- |
| Patients diagnosed with:<br>• severe or mild rheumatic valve disease<br>• severe or mild calcific aortic valve stenosis<br> (As per Jones criteria 2015, WHF, and 2020 ACC/AHA guidelines) | Patients with:<br>• autoimmune inflammatory disorders<br>• atherosclerosis<br>• HIV infection,<br>• cardiomyopathies,<br>• congenital valve lesions<br>• previous cardiac surgery or valve repair |
| Patients aged 18–70 years | Patients aged <18 or >70 years |
| Patients with written informed consent | Patients unable to consent |

WHF; World Heart Federation, ACC/AHA; American College of Cardiology/American Heart Association, HIV; human immunodeficiency virus.

The cadaveric tissues will be considered healthy if they meet the inclusion and exclusion criteria (Table 2) [44, 45].

## Research procedures

Patients and living controls will undergo a thorough clinical assessment after signing informed consent. Blood samples will be collected from patients with mild and severe VHD recruited at the Cardiac Clinic, Groote Schuur Hospital, Cape Town, and compared to samples from matched controls. Furthermore, explanted valves at valve replacement surgery, ventricular biopsies, atrial biopsies, and serum, plasma, and pericardial fluid samples will be obtained from severe VHD patients. Nondiseased cardiac tissue biopsies will be collected from the cadavers of persons who died due to trauma-related injuries requiring planned full autopsies at the Salt River Forensic Pathology Services Mortuary in Cape Town. To recruit cadaveric tissue donors, informed consent will be obtained from the next of kin or legal guardians. Cardiovascular imaging assessments will be performed on mild and severe VHD patients and controls using CMR and TTE (Fig 1).

## Clinical assessments

Participants' demographics, clinical history, cardiac function, and diagnostic laboratory variables will be recorded in case reference forms (CRFs) specifically designed for this study and

**Table 2. Inclusion criteria for cadaveric tissue donors.**

| |
| --- |
| No evidence of previous cardiac surgery |
| No history of heart disease. |
| No left ventricular hypertrophy (LV wall thickness <15 mm measured 2 cm below the mitral valve). |
| Heart weight within normal range (men:212–373 g; women:164–317 g) [43]. |
| Coronary artery (CA) atherosclerosis occluding <50% luminal surface area in right CA, left mainstem CA, left anterior descending CA, or circumflex artery. |
| No macroscopic evidence of previous myocardial infarction (fibrosis). |
| Cardiac valve circumference within reference ranges [42]. |
| No thickening, calcification or fusion of valve cusps or chordae tendinae. |
| No evidence of current or previous pericarditis (adhesions). |
| No history or observation of sepsis at autopsy. |
| No drug toxicity related death. |

reviewed by the Human Research Ethics Committee. Medical records will be reviewed for all the recruited VHD patients to obtain a detailed history, cardiovascular risk factors (comorbidities), and therapies prescribed. Ventricular, atrial, and valvular echocardiography measurements will be obtained from the medical records of consenting VHD patients. The diagnosis and grading of valve lesions will be performed as per the WHF Revised Jones Criteria (2015) and ACC/AHA valvular heart disease diagnosis guidelines (2020) [4, 8, 41]. The TTE variables will be obtained from consenting VHD patients and healthy individuals recruited into the study. Diagnostic biochemical results obtained during routine clinical care will be reviewed and recorded from the South African National Health Laboratory Services platform. Heart weight, time since death, age, ethnicity, and sex will be recorded for cadaveric donors. The measured variables will be compared between RHD patients, AS patients, and controls using ANOVA or the Kruskal–Wallis H test for numeric variables and Pearson's chi-square ($\chi^2$) test for categorical variables. The variables will be considered significantly different if they meet the p value cut-off ($p<0.05$).

## Cardiovascular assessment

**Transthoracic echocardiography.** TTE will be performed on all patients in the left lateral position by experienced sonographers using an S5-1 transducer on a GE Vivid S6 system (Horten, Norway). Images will be acquired according to a standardised protocol. The data will be transferred and analysed off-line using the Xcelera workstation. All linear chamber measurements will be performed according to the American Society of Echocardiography (ASE) chamber guidelines. Measurements relating to left ventricle (LV) diastolic function will be performed in accordance with the ASE guidelines on diastolic function and will include pulse-wave Doppler at the mitral tips and tissue Doppler of both the medial and lateral mitral annuli. The diagnosis of RHD and degenerative AS will be based on the WHF (2015) and ACC/AHA 2020 valvular heart disease diagnosis guidelines. In the case of ambiguity, the echocardiographic data will be integrated with the clinical evaluation by an experienced cardiologist to confirm the diagnosis.

During the study, the following TTE parameters will be collected to assess ventricular function and valve pathologies. The left ventricular study will include the LV ejection fraction, systolic and diastolic volumes, end diastolic diameter and LV mass. The assessment of the aortic valve (AV) will include the AV area, velocity and gradients across the valve, left atrial area, aortic root area and LV outflow tract velocity and diameter. The mitral valve (MV) variables will include the MV area, gradient, and peak velocity for the calculation of the E/A and E/e' ratios to assess diastolic function and deceleration time. Finally, aortic and mitral valve morphologies will be described. The measured TTE variables will be statistically compared between the RHD, degenerative aortic stenosis, and control groups. The variables will be compared using ANOVA or the Kruskal–Wallis H test for numeric variables and Pearson's chi–square ($\chi^2$) test for categorical variables. Differences will be considered significant if they meet the p value cut-off ($p<0.05$).

**Cardiovascular magnetic resonance imaging.** CMR studies will be performed on a 3-Tesla scanner (Siemens Healthcare, Erlagen, Germany) using an 18-channel phased-array body coil. The images will be obtained during patient expiratory breath-holding for approximately six seconds and will be prospectively ECG gated. LV volumes and mass will be acquired in line with standard institutional CMR protocols. Balanced steady-state-free-precession (b-SSFP) imaging (repetition time = 43.08 ms, echo time = 1.61 ms, flip angle = 40 degrees, matrix size = 149 × 208, bandwidth = 962 Hz/Px, slice = 8 mm thickness, 25 phases) will be performed to obtain long-axis cines and contiguous short-axis stack cines for the assessment of LV

volume, mass, and ejection fraction, acquired over 9 heartbeats per slice. CMR cine images will be acquired in short-axis stacks at the base, middle and apex of the heart to evaluate myocardial deformation using feature tracking analysis. Myocardial deformation will be assessed by calculating circumferential, radial, and longitudinal strain and strain rates from cine images. Parametric mapping allows direct quantification of T1 and T2 relaxation times and the extracellular volume fraction (ECV) of the myocardial tissue by providing tissue-specific T1 and T2 time values and allowing comparison of quantified myocardial parameters with normal reference values obtained from the scanner. Each ECV measurement will be obtained by subtracting pre- and postcontrast maps with haematocrit correction, acquired approximately 15 minutes after the administration of contrast.

Late gadolinium enhancement (LGE) imaging will be performed 6–15 minutes after gadolinium administration, and images will be acquired using a short-axis stack, 2-chamber, 4-chamber, and LV outflow tract (LVOT) images to assess focal myocardial fibrosis. Images will be analysed by independent, experienced readers who are blinded to the echocardiographic results and who have more than 5 years of experience using CVI42® software from Circle Cardiovascular Imaging (version 5.11.2, Calgary, Alberta, Canada). The assessment of cardiac function and chamber sizes will be performed in standard views in the long-axis (horizontal and vertical) and short-axis planes. To assess the valve flow dynamics, flow images will be acquired using velocity-encoded phase contrast imaging of the mitral, aortic, tricuspid, and pulmonary valves. Velocity-encoded images will be acquired within 0–6 minutes of contrast agent administration. Furthermore, to study the aortic valves, the following measurements will be recorded: forward flow, backwards or reverse flow, maximum velocity, regurgitant fraction, pressure gradient (Max Aortic PG), aortic root anatomy, aortic wall pressure, and aortic valve area. To study the mitral valves, the following parameters will be recorded: regurgitant volume and fraction, diastolic flow, and velocity across the mitral valves, MV area, and mitral valve leaflet morphology. To study the pulmonary valves, the following parameters will be obtained: forward flow (PA FFlow), backwards or reverse flow (PA Rflow), maximum velocity (PA Max Vel), regurgitant fraction (RF %), and pressure gradient (Max Pulm PG). The variables will be compared among RHD patients, patients with degenerative aortic stenosis, and controls using ANOVA or the Kruskal–Wallis H test for numeric variables and Pearson's chi-square ($\chi^2$) test for categorical variables. Differences will be considered significant if they meet the p value cutoff ($p < 0.05$).

## Blood and tissue sample collection and processing

Venous blood will be collected from RHD patients, AS patients and healthy controls into 2 VACUETTE® Z serum clot activator vacutainer 6 mL tubes (Greiner Bio-One International GmbH, Kremsmünster, Austria) for serum isolation and into 3 VACUETTE® K3E K3EDTA vacutainer 9 mL tubes (Greiner Bio-One International GmbH, Kremsmünster, Austria) for plasma isolation. If possible, all consumables will be bought at the beginning of the study as a single batch to minimise lot-to-lot differences [46]. The blood samples will be kept at room temperature for up to 40 minutes before processing. Plasma and serum will be isolated by centrifuging the blood samples at $2,000 \times g$ for 15 minutes in a cooled centrifuge (4°C), after which 500 µL aliquots of plasma and serum will be obtained and stored at -80°C until analysis. Platelet-free plasma for miRNA extraction will be obtained by respinning 1.5 mL aliquots of plasma in 2 mL conical Eppendorf tubes for 10 minutes at $16,000 \times g$ in a fixed-angle rotor centrifuge cooled to 4°C. The platelet-free plasma will be aliquoted into 250 µL aliquots, stored in RNase-free microtubes and stored at -80°C until analysis. Valves, right atrium, and LV biopsies will be collected after excision by the cardiothoracic surgeon or forensic pathologist.

The left ventricle biopsies from the consenting patients will be obtained using 14G x 7.6 cm Tru-CutTM biopsy needles (Becton, Dickinson U.K. Limited, Berkshire, England). The valve biopsies will be immediately transversely dissected perpendicular to the cut edge (valve ring) and free edge of the cusps and aliquoted. The aliquots used for H&E staining will be immersed in 10% neutral buffered formalin (NBF), while the other aliquots will be immediately snap frozen in liquid nitrogen and stored at -80˚C until analysis. The H&E-stained aliquots will be stored in 10% NBF until they are embedded in wax, and the blocks will be carefully archived until histology experiments. Furthermore, whole blood and plasma will be collected on Whatman® filter paper to prepare dried blood and dried plasma spots for pilot experiments. The dried spots will be stored either at room temperature, in a cold 4˚C room, or in a -80˚C freezer and analysed to determine signal stability over time. Humidity indicator cards will be used to monitor humidity on the dried spot filters (Fig 1).

## Laboratory experiments

Valve fragments and blood samples will be immediately processed into plasma and serum and stored at -80˚C until analysis. The experiments will be conducted at the Cape Heart Institute, Department of Medicine; Division of Molecular Pathology, Department of Pathology; Pharmacology and Toxicology Laboratory, Division of Clinical Pharmacology, Department of Medicine; Division of Forensic Toxicology, Department of Pathology; and the Hair and Skin Research Laboratory, Department of Medicine, University of Cape Town (Fig 1).

**Assessment of immunohistology and autophagy biomarkers.** H&E-stained thin sections will be used for histopathological assessment to confirm cases of chronic RHD or chronic degenerative AS. Sections will be reviewed by an experienced pathologist for histological features of RHD or degenerative AS [47–49]. To characterise the involvement of autophagy in the pathogenesis of RHD and degenerative AS, the expression of Beclin, microtubule-associated protein 1A/1B-light chain 3 (LC3), p62, bcl-2-associated X protein (BAX), Bcl-2 and caspase-3 in patients will be analysed using immunohistochemistry. Valve tissue samples will be processed using a Leica tissue processing machine (Leica TP 1020; Leica Microsystems, Nussloch, Germany). Valve sections will be cut and prepared for immunoblotting with antibodies against LC3B, Beclin 1 (BECN1), BAX, Bcl2, and caspase-3. Antibody detection will be performed with a DAKO envision kit (Leica Biosystems, DS9800) following the manufacturer's instructions. The stroma and macrophages will be detected by assessing the intensity, proportion and number of stroma and macrophages. Some pilot experiments will be conducted to optimise the immunohistochemistry tests. The distribution of the measured variables on the valve biopsies will be compared between the RHD and degenerative aortic stenosis groups using chi-square or Fisher's exact tests. The experiments will be performed in triplicate.

**Western blots and mass spectrometry proteomics.** To assess the integrity of tissues obtained from cadaveric donors, western blotting will be used. Following sampling, heart tissue will be homogenised under dry liquid nitrogen, and proteins will be extracted and quantified by a bicinchoninic acid protein estimation assay. Sodium dodecyl sulfate–polyacrylamide gel electrophoresis (SDS–PAGE) and western blotting will be used to analyse protein expression patterns in cadavers at different postmortem intervals (PMIs). The expression patterns of the cardiac troponin I/T, desmin, tropomyosin, glycerol-3-phosphate dehydrogenase (GPDH), talin 1, Eef1a and vinculin proteins will be analysed in LV and valve biopsies. ImageJ software will be used for quantitative analysis of the expression patterns. The postmortem intervals will be grouped into intervals of 4 hours. The western blot bands will be quantified using ImageJ (https://imagej.net/ij/index.html) to determine the normalised intensities per sample. The intensities of the analysed proteins will be measured and compared between the

grouped postmortem intervals using parametric or nonparametric hypothesis testing techniques. The differences between the groups will be measured using univariate statistical methods (p value cut-off < 0.05). The western blot experiments will be performed in triplicate.

To characterise the RHD and AS circulatory and tissue-specific proteomes, untargeted liquid chromatography tandem mass spectrometry (LC–MS/MS) proteomics will be conducted. The entire proteome will be extracted from plasma/serum and valve fragments, fractionated, digested, and analysed with a Q Exactive mass spectrometer [50]. The data will be processed and analysed with MaxQuant and Perseus [51, 52]. The differential expression of significant proteins between RHD patients, patients with degenerative AS, and healthy controls will be compared. To further associate the significant proteins with the valve pathologies associated with RHD and degenerative AS, a semitargeted proteomics strategy will be utilised to colocalise the significant peptides with observed valve pathologies such as calcification, fibrosis, collagenisation, inflammation, and hyalinization using matrix-assisted laser desorption/ionisation imaging mass spectrometry (MALDI-MSI) [53, 54]. Pilot experiments will be performed to establish suitable parameters for data acquisition, data processing, and analysis. The significant proteomics biomarkers will be obtained after comparing the expression between the groups. Proteins with a fold change greater than ±2 and a significant p value (p< 0.05) will be considered potential biomarkers. The *m/z* features detected in the MALDI-MSI experiments with an AUC-ROC>0.7 will be considered important for discriminating the compared regions of interest (ROIs) on the analysed slides. The MALDI experiments will be performed in duplicate by mouting 2 successive same-tissue sections side by side.

**Circulatory and tissue-specific metabolomics.**   To further explore the molecular profiles of RHD patients, degenerative AS patients and healthy controls, untargeted LC–MS/MS metabolomics will be performed on plasma/serum and valve biopsies from patients with severe valve diseases. In addition, dried whole blood and dried plasma spot pilot experiments will be performed to determine the stability of metabolites in dried filter papers compared to that in liquid plasma. Pilot experiments will determine the suitability of using dry spots as alternative sample collection methods, especially in remote regions of SSA. To characterise the metabolome in patients with valve diseases, blood samples, dried spots, and valve biopsies will be analysed with liquid chromatography quadrupole time-of-flight tandem mass spectrometry (LC Q-TOF-MS/MS), and the data will be processed with MS-DIAL, annotations will be performed with Global Natural Products Social Molecular Networking (GNPS) software, and statistical analysis will be performed using MetaboAnalyst [55–57]. To explore tissue-specific and circulatory metabolomics profiles, the expression levels of significant metabolites will be compared between blood samples and valve fragments from patients with severe valve diseases. Furthermore, the spatial localisation and correlation of the dysregulated metabolites to valve pathologies will be explored with MALDI-MSI [58]. Statistical analyses of metabolomics data will be conducted using both univariate (Wilcoxon and Kruskall Myer) and multivariate (PCA, random forest) methods. The data will initially be treated as nonnormal, but if they are normally distributed, parametric methods will also be introduced. In bulk metabolomics experiments (i.e., LC–MS), potential metabolic biomarkers will be identified as metabolites with a variability of at least 20% greater than the technical variability and a significant p value (p<0.05) between the studied groups. For metabolic imaging, *m/z* features are defined as features of interest if they are capable of differentiating between regions of interest with an AUC-ROC greater than 0.7. Samples for the MALDI experiments will be analysed in duplicate by mouting 2 successive same-tissue sections side by side.

**Blood and tissue transcriptome analyses.**   To describe the transcriptome profile of patients with VHD, microRNAs will be extracted from platelet-free plasma and frozen valve fragments [59, 60]. The miRNeasy Serum/Plasma and miRNeasy Tissue/Cells kits will be used

for miRNA extraction according to the manufacturers' protocols (Qiagen, Hilden, Germany). Targeted and untargeted transcriptome analysis will be used to quantify dysregulated miRNAs [59–62]. High-throughput techniques will be used to sequence the extracted miRNAs. Integrative bioinformatics will be used to determine the mRNAs targeted by the altered miRNAs in RHD and AS and correlate them to proteomics and metabolomics biomarkers [59–62]. Transcriptomic biomarkers with a significant q value (q <0.05 will be considered important.

## Multiomics integration analysis

To determine the interconnectivity of dysregulated biomarkers, data integration methods will be used to analyse the omics data [52, 56]. To determine the biological relevance of the dysregulated molecules, functional analyses will be conducted on the proteomics, metabolomics, and transcriptomics results using bioinformatics tools. Furthermore, multiomics analyses will be conducted to characterise protein–protein interactions, protein-metabolite interactions, protein–gene interactions and metabolite-gene interactions using systems biology tools to provide an in-depth understanding of the molecular processes involved in RHD and degenerative AS [56, 63, 64].

## Impact of the findings, hypothesis, and limitations

Previous studies have reported on the role of molecular biomarkers in understanding the pathogenesis and etiology of degenerative aortic valve diseases. Some proteomic and metabolomic studies have identified potential biomarkers associated with worsening valve pathology and cardiac output parameters [14, 16, 18, 19]. However, most of the biomarkers reported are based on studies conducted in high-income countries, with few focusing on low- and middle-income countries (LMICs). Patients residing in resource-limited countries (RLCs) in Africa have diverse diets, lifestyles and ethnicities; therefore, locally generated data are extremely important for better diagnosis and prognosis in these populations. The incidence of RHD in particular is much greater in LMICs than in high-income countries, and the increasing incidence of degenerative calcific aortic stenosis complicates the diagnosis and management of valvular heart disease patients [3]. Few studies have explored biomarkers to understand the pathomechanisms of rheumatic heart disease [10, 13, 22, 24, 39]. Therefore, we hypothesise that our proposed study will report on common and differential biomarkers associated with degenerative aortic stenosis and rheumatic heart disease in the African population. We also aim to determine the reproducibility and applicability of the reported biomarkers by validating them in other African cohorts. In addition, we will report on the factors that may influence the suitability of using cadaveric tissues as controls in cardiovascular disease research, enabling this study to serve as a proof of concept for the use of cadaveric cardiac tissues for biomarker research. Finally, we hypothesise that our study will raise awareness in the basic sciences community of the need for further molecular biology studies to advance our understanding of the pathomechanisms related to underresearched cardiovascular diseases in Africa. We foresee some challenges and limitations in the study. It will be challenging to acquire healthy control tissue biopsies from healthy donor cadavers immediately after they are declared deceased. Therefore, we have planned pilot studies to determine suitable postmortem intervals for inclusion as controls. Furthermore, we foresee a challenge in the recruitment rate of patients undergoing valve replacement surgery. Inability to ascribe causality of the omics biomarkers to the observed pathologies is a potential limitation of the study. However, the longitudinal follow-up of the participants and the use of a multiomics approach might shed some light on the impact that the reported biomarkers have on the progression of valvular heart disease pathologies.

## Patient and public involvement

During the development of the research questions and design of the study, the public and the patients were not involved. However, the findings from the study will be submitted for publication in peer-reviewed journals and presented in conference meetings. Findings that may affect patient management will be presented to patients and advocacy groups through public engagement.

## Study status

The study started in 2018 and we have currently recruited 80 VHD patients and 40 healthy participants. VHD patients and healthy participants were characterised by TTE and CMR, blood sample collection, and tissue biopsy collected from patients undergoing valve replacement surgery. Participant recruitment is still ongoing, and a pilot study to collect tissue from deceased individuals is ongoing. The study is expected to be concluded in December 2026.

## Ethics, dissemination, and data availability

The study received ethical approval from the UCT Human Research Ethics Committee (HREC REF: 554/2017, 061/2018, 574/2018, and 674/2020). Written and duly signed informed consent will be obtained from all study participants; thumb-printed consent duly signed by witnesses in the case of illiterate participants or relatives will also be obtained. In addition, family consent for postmortem tissue donation will be obtained as described by Chapter 8 of the South African National Health Act 61 of 2003 as well as internal guidelines [65]. The study will strictly adhere to the principles of the Helsinki Declaration of 2013 [66]. The clinical data of the participants will be anonymised with a unique identifier stored in a password-locked database, and the source files will be stored in a secure locker. The samples will be stored as per approved standard operating procedures and good clinical practice guidelines. The study has a data management plan that prescribes that a data transfer agreement will be processed before any data can be transferred to a third party. Primary and secondary findings from the study will be published in peer-reviewed journals. After the conclusion of the study and the publication of the findings, the data will be made fully available, as far as legally allowable, through ZivaHub (https://zivahub.uct.ac.za/), which is a searchable and publicly available data repository managed and maintained by the University of Cape Town. The clinical metadata will be anonymised before being made publicly available.

## Discussion

RHD is endemic in LMICs and SSA. The prevalence of degenerative AS is on the rise in SSA due to the increasing prevalence of ASCVD risk factors (diabetes, hypertension, smoking, dyslipidemia, and obesity). The confluence of the two valvular heart diseases in Africa puts strain on the existing healthcare systems. Therefore, our study aims to describe the pathophysiological imaging markers in RHD and AS patients. Furthermore, we intend to investigate the pathogenesis of RHD in contrast to degenerative AS and identify molecular markers for early detection and diagnosis to determine the molecular pathomechanisms involved and to identify potential therapeutic targets.

Several studies have investigated the pathogenesis of RHD and degenerative AS and have elucidated the pathomechanisms associated with RHD and AS [10, 13, 31, 67, 68]. However, there is a paucity of data on imaging and molecular biomarkers in patients with early-stage and late-stage RHD, especially in ARF-endemic regions [6, 10, 13]. To the best of our knowledge, even fewer studies have compared the molecular mechanisms of RHD and AS.

Proteomic studies on AS and RHD have led to an enhanced understanding of the molecular mechanisms leading to the valve pathologies observed in AS and RHD patients [31]. However, very few metabolomics studies have investigated the metabolic pathomechanisms involved in RHD [39].

The use of CMR, proteomics and transcriptomics has had a considerable impact on disease characterisation, diagnosis, and understanding of the disease processes in VHD [31, 69, 70]. The use of multiple techniques and platforms to understand VHD provides a unique opportunity to describe robust biomarkers that explain mechanistic disease progression [31, 70].

Furthermore, tissue-specific biomarker experiments in CVD patients are limited due to challenges in accessing surgically obtained valve biopsies from patients [31], but a greater challenge is obtaining nondiseased tissue biopsies as controls [40]. A handful of studies have used cadaveric samples as controls, and even fewer have used this strategy in the context of CVD [71–73]. None of these studies have discussed their approach to an acceptable PMI for use in their experiments. Our study will provide important insights into the selection of cadaveric samples as controls in cardiovascular studies utilising proteomics and metabolomics. In multi-center studies, sample handling and transportation to biobanks or laboratories for analysis are the main causes of variation in the results [74–76]. The variations are astronomical when using sensitive techniques such as metabolomics; however, the use of dried blood spot analysis has been shown to be stable and does not require special handling during transport [77]. This study explored the suitability of using dried whole blood or dried plasma spot techniques/platforms to collect samples for metabolomics studies, especially in remote regions with limited refrigeration facilities, such as many LMICs.

This study will not only provide imaging and molecular biomarkers for the early detection and diagnosis of RHD and AS but also provide insight into suitable cadaveric samples for use as controls. In addition, this study will provide an alternative method for sample handling and storage suitable for resource-limited regions in Africa.

## Acknowledgments

The authors would like to thank Prof. Dhiren Govender, Riyaadh Roberts, and Subash Govender for H&E sectioning, staining, and slide review; Alicia Evans for LCMS/MS facility support and training; and Prof. Nonhlanhla Khumalo and Nandi Mehlala for MALDI-MSI facility access and technical support.

## Author Contributions

**Conceptualization:** Sebastian Skatulla, Ntobeko A. B. Ntusi.

**Data curation:** Daniel W. Mutithu, Olukayode O. Aremu, Dipolelo Mokaila, Tasnim Bana, Laura Taylor, Lorna J. Martin, Laura J. Heathfield, Jennifer A. Kirwan, Lubbe Wiesner, Henry A. Adeola, Evelyn N. Lumngwena, Sebastian Skatulla, Richard Naidoo, Ntobeko A. B. Ntusi.

**Formal analysis:** Daniel W. Mutithu, Olukayode O. Aremu.

**Funding acquisition:** Sebastian Skatulla, Ntobeko A. B. Ntusi.

**Investigation:** Daniel W. Mutithu, Dipolelo Mokaila, Tasnim Bana, Mary Familusi, Laura Taylor, Lorna J. Martin, Jennifer A. Kirwan, Lubbe Wiesner, Henry A. Adeola, Evelyn N. Lumngwena, Rodgers Manganyi, Richard Naidoo, Ntobeko A. B. Ntusi.

**Methodology:** Daniel W. Mutithu, Olukayode O. Aremu, Laura J. Heathfield, Jennifer A. Kirwan, Lubbe Wiesner, Henry A. Adeola, Evelyn N. Lumngwena, Sebastian Skatulla, Richard Naidoo, Ntobeko A. B. Ntusi.

**Project administration:** Ntobeko A. B. Ntusi.

**Resources:** Ntobeko A. B. Ntusi.

**Supervision:** Laura Taylor, Jennifer A. Kirwan, Lubbe Wiesner, Henry A. Adeola, Sebastian Skatulla, Richard Naidoo, Ntobeko A. B. Ntusi.

**Validation:** Tasnim Bana, Ntobeko A. B. Ntusi.

**Writing – original draft:** Daniel W. Mutithu.

**Writing – review & editing:** Olukayode O. Aremu, Dipolelo Mokaila, Tasnim Bana, Mary Familusi, Laura Taylor, Lorna J. Martin, Laura J. Heathfield, Jennifer A. Kirwan, Lubbe Wiesner, Henry A. Adeola, Evelyn N. Lumngwena, Rodgers Manganyi, Sebastian Skatulla, Richard Naidoo, Ntobeko A. B. Ntusi.

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
