## [Decision Letter · Decision Letter 0]

13 Mar 2024

PONE-D-24-00937A study protocol to characterise pathophysiological and molecular markers of rheumatic heart disease and degenerative aortic stenosis using multiparametric cardiovascular imaging and multi-omics techniquesPLOS ONE

Dear Dr. Mutithu,

Thank you for submitting your manuscript to PLOS ONE. After careful consideration, we feel that it has merit but does not fully meet PLOS ONE’s publication criteria as it currently stands. Therefore, we invite you to submit a revised version of the manuscript that addresses the points raised during the review process.

We look forward to receiving your revised manuscript.

Kind regards,

Mohanad Alkhodari

Academic Editor

PLOS ONE

Comments from Senior Staff Editor: We note that one or more reviewers has recommended that you cite specific previously published works. As always, we recommend that you please review and evaluate the requested works to determine whether they are relevant and should be cited. It is not a requirement to cite these works. We appreciate your attention to this request.

3. We note that Figures 1, 2 and 3 in your submission contain copyrighted images. All PLOS content is published under the Creative Commons Attribution License (CC BY 4.0), which means that the manuscript, images, and Supporting Information files will be freely available online, and any third party is permitted to access, download, copy, distribute, and use these materials in any way, even commercially, with proper attribution. For more information, see our copyright guidelines: http://journals.plos.org/plosone/s/licenses-and-copyright.

1. You may seek permission from the original copyright holder of Figures 1, 2 and 3 to publish the content specifically under the CC BY 4.0 license.

Reviewers' comments:

Reviewer's Responses to Questions

**Comments to the Author**

1. Does the manuscript provide a valid rationale for the proposed study, with clearly identified and justified research questions?

Reviewer #1: Yes

Reviewer #2: Yes

2. Is the protocol technically sound and planned in a manner that will lead to a meaningful outcome and allow testing the stated hypotheses?

Reviewer #1: No

Reviewer #2: Yes

3. Is the methodology feasible and described in sufficient detail to allow the work to be replicable?

Reviewer #1: No

Reviewer #2: Yes

4. Have the authors described where all data underlying the findings will be made available when the study is complete?

Reviewer #1: No

Reviewer #2: Yes

5. Is the manuscript presented in an intelligible fashion and written in standard English?

Reviewer #1: Yes

Reviewer #2: Yes

6. Review Comments to the Author

You may also provide optional suggestions and comments to authors that they might find helpful in planning their study.

Reviewer #1: In the manuscript titled "A Study Protocol to Characterize Pathophysiological and Molecular Markers of Rheumatic Heart Disease and Degenerative Aortic Stenosis Using Multiparametric Cardiovascular Imaging and Multi-omics Techniques," the authors outline a protocol for patient recruitment and the rationale for conducting the study. This initiative is commendable, especially in light of the latest recommendations for open science, which emphasize the importance of publishing study protocols before conducting research and subsequently sharing the results. However, several significant modifications are necessary before publication and, crucially, before initiating the study.

1. **Inclusion Criteria:** The specified age range of 18-80 years is overly broad, considering the significant physiological differences across younger, middle-aged, and older patients. This criterion should be reassessed, or at the very least, patients should be categorized into distinct age groups as recommended, ensuring adequate representation in each category.

2. **Sex Differences:** The protocol currently overlooks the critical aspect of sex differences in cardiovascular disease (CVD) pathology. Given the historical underrepresentation of females in CVD research, it is imperative to aim for gender-balanced groups to address this gap effectively.

3. **Sample Size:** The document does not specify the number of participants for each group, a critical omission. While calculating the power of multi-omics studies may be challenging, it is still necessary to provide power calculations or at least an estimate of the minimum number of subjects required for each omics approach to ensure reliable results.

4. **Sample Collection and Processing:** Detailed information on blood and tissue sample collection and processing is lacking. This should include the types and brands of collection tubes, the number of tubes for serum/plasma, tube volume, anticoagulants used, centrifugation speed and conditions for serum/plasma separation, time from collection to storage, aliquot numbers, and storage conditions. For transcriptomics, considering the variability in platelet counts among patients, which can significantly affect results, platelet-poor plasma may be preferable for miRNA analysis. Details on RNA preservation and extraction methods for both serum/plasma and tissue samples, including the use of RNA later, Trizol, or snap freezing with liquid nitrogen, should be provided.

5. **Open Research Data Initiative and FAIR Principles:** The manuscript should also address the Open Research Data Initiative and incorporate the FAIR (Findable, Accessible, Interoperable, and Reusable) principles. Discussing how the study's data will be made available in accordance with these principles is essential. This includes detailing the data management plan, which should outline how data will be stored, shared, and made accessible to ensure that it is discoverable and usable by others in the research community. Highlighting the commitment to these practices not only aligns with the push towards open science but also enhances the transparency, reproducibility, and impact of the research.

The considerations and suggestions from the referenced paper (https://doi.org/10.1016/j.molmed.2023.09.004) provide valuable guidance on designing and conducting multi-omic studies. While direct citation is not needed, employing this resource as a framework for study design and execution can greatly enhance the protocol's robustness and scientific contribution.

Reviewer #2: The current study represent an interesting topic to study and to conduct. We have few comments:

1- There is a need to further support the current proposed sample size (120 pts).

2- Clarify properly if you recruit the patients from endemic area of RHD or not. The current status of RHD.

3- Justify how your cohort would represent a real world cohort regarding RHD and AS.

4- High light your measures to avoid bias, selection bias.

5- Discuss the expected major limitations or challenges.

Regards

7. PLOS authors have the option to publish the peer review history of their article (what does this mean?). If published, this will include your full peer review and any attached files.

Reviewer #1: No

Reviewer #2: **Yes: **Rami Riziq Yousef Abumuaileq

---

## [Author Response · Author response to Decision Letter 0]

11 Apr 2024

Reviewer's Responses to Questions

Comments to the Author

1. Does the manuscript provide a valid rationale for the proposed study, with clearly identified and justified research questions?

Reviewer #1: Yes

Reviewer #2: Yes

2. Is the protocol technically sound and planned in a manner that will lead to a meaningful outcome and allow testing the stated hypotheses?

Reviewer #1: No

Reviewer #2: Yes

Response: Thank you for the feedback. In the revised manuscript we have included the specific descriptions of the cutoff points and the specific experiments to be conducted.

3. Is the methodology feasible and described in sufficient detail to allow the work to be replicable?

Reviewer #1: No

Reviewer #2: Yes

Response: Thank you for the feedback. In the revised manuscript we have included specifics of the experiments to be conducted and their replications where appropriate. We have also included the accepted power and sample size calculations.

4. Have the authors described where all data underlying the findings will be made available when the study is complete?

Reviewer #1: No

Reviewer #2: Yes

Response: We have included a data availability statement. Lines 519 – 521 of the revised manuscript. 

5. Is the manuscript presented in an intelligible fashion and written in standard English?

Reviewer #1: Yes

Reviewer #2: Yes

Response: The revised clean manuscript has been reviewed, proofread, and the language edited.

6. Review Comments to the Author

You may also provide optional suggestions and comments to authors that they might find helpful in planning their study.

Reviewer #1: In the manuscript titled "A Study Protocol to Characterize Pathophysiological and Molecular Markers of Rheumatic Heart Disease and Degenerative Aortic Stenosis Using Multiparametric Cardiovascular Imaging and Multi-omics Techniques," the authors outline a protocol for patient recruitment and the rationale for conducting the study. This initiative is commendable, especially in light of the latest recommendations for open science, which emphasize the importance of publishing study protocols before conducting research and subsequently sharing the results. However, several significant modifications are necessary before publication and, crucially, before initiating the study.

Response: Thank you for taking the time to review our manuscript and for your kind and constructive feedback.

1. **Inclusion Criteria:** The specified age range of 18-80 years is overly broad, considering the significant physiological differences across younger, middle-aged, and older patients. This criterion should be reassessed, or at the very least, patients should be categorized into distinct age groups as recommended, ensuring adequate representation in each category.

Responses: Thank you for the feedback. After careful consideration, we have revised the upper age limit to 70 years. The mean age of the patients currently recruited in the study is 44.7±13.7 years for RHD patients and 64.2±12.8 years for degenerative AS patients. We acknowledge that the range is larger than desirable, especially when age is a frequent confounder in such studies. However, degenerative AS is often related to advanced age. Furthermore, at the location where we are recruiting the participants, most of the patients present for valve replacement at the advanced stages of the disease (both RHD and AS) which also tends to make them older than in other populations.

Given the larger sample sizes we are now looking to recruit, we believe that there should be sufficient age stratification to enable a confounding factor of age to be appropriately accounted for.

2. **Sex Differences:** The protocol currently overlooks the critical aspect of sex differences in cardiovascular disease (CVD) pathology. Given the historical underrepresentation of females in CVD research, it is imperative to aim for gender-balanced groups to address this gap effectively.

Response: Thank you for highlighting this information. We have added a statement indicating that we shall ensure that there is a balance between males and females in the studied groups. Lines 260-261.

3. **Sample Size:** The document does not specify the number of participants for each group, a critical omission. While calculating the power of multi-omics studies may be challenging, it is still necessary to provide power calculations or at least an estimate of the minimum number of subjects required for each omics approach to ensure reliable results.

Response: Thank you for your suggestions. After reconsideration and analysis, we have increased the sample size to 250. This is based on the information obtained from using available tools for power calculation for metabolomics and genomics studies. In the revised manuscript we have included the justification for using a sample size of 250 VHD participants and an equal number of controls. We have also broken down the sample size per disease group and categories.

4. **Sample Collection and Processing:** Detailed information on blood and tissue sample collection and processing is lacking. This should include the types and brands of collection tubes, the number of tubes for serum/plasma, tube volume, anticoagulants used, centrifugation speed and conditions for serum/plasma separation, time from collection to storage, aliquot numbers, and storage conditions. For transcriptomics, considering the variability in platelet counts among patients, which can significantly affect results, platelet-poor plasma may be preferable for miRNA analysis. Details on RNA preservation and extraction methods for both serum/plasma and tissue samples, including the use of RNA later, Trizol, or snap freezing with liquid nitrogen, should be provided.

Response: In the revised manuscript we have added the detailed information for sample collection and processing. We have also added some detailed information on miRNA extraction and handling.

5. **Open Research Data Initiative and FAIR Principles:** The manuscript should also address the Open Research Data Initiative and incorporate the FAIR (Findable, Accessible, Interoperable, and Reusable) principles. Discussing how the study's data will be made available in accordance with these principles is essential. This includes detailing the data management plan, which should outline how data will be stored, shared, and made accessible to ensure that it is discoverable and usable by others in the research community. Highlighting the commitment to these practices not only aligns with the push towards open science but also enhances the transparency, reproducibility, and impact of the research.

Response: We have added a data availability statement in the revised manuscript. Lines 548-550.

The considerations and suggestions from the referenced paper (https://doi.org/10.1016/j.molmed.2023.09.004) provide valuable guidance on designing and conducting multi-omic studies. While direct citation is not needed, employing this resource as a framework for study design and execution can greatly enhance the protocol's robustness and scientific contribution.

Response: Thank you for the suggestion. The article was very informative.

Reviewer #2: The current study represent an interesting topic to study and to conduct.

Response: Thank you for taking your time to review our manuscript and for your kind and positive feedback.

 We have few comments:

1- There is a need to further support the current proposed sample size (120 pts).

Response: Thank you for the feedback. We have increased the sample size to 250 participants, and we have also included a justification for the sample size in the revised manuscript. 

2- Clarify properly if you recruit the patients from endemic area of RHD or not. The current status of RHD.

Response: We are recruiting in RHD endemic area, the current prevalence of RHD is 29.40/100,000 as at 2017.

3- Justify how your cohort would represent a real-world cohort regarding RHD and AS.

Response: We are a little confused by your answer/request. This population represents a real-world cohort in that it is reflective of the patients that we are regularly treating daily in Cape Town hospitals. In this respect, being able to distinguish between AS and RHD where the prevalence of both is high becomes an important clinical question.

4- High light your measures to avoid bias, selection bias.

Response: Since this is a case-control study we have defined clear inclusion criteria that will be used to recruit patients into RHD or AS group and the healthy controls.

5- Discuss the expected major limitations or challenges.

Response: Thank you for the suggestion. In the revised manuscript we have highlighted some potential limitations and challenges to the study. Lines 524 – 531.

Regards

7. PLOS authors have the option to publish the peer review history of their article (what does this mean?). If published, this will include your full peer review and any attached files.

Do you want your identity to be public for this peer review? For information about this choice, including consent withdrawal, please see our Privacy Policy.

Reviewer #1: No

Reviewer #2: Yes: Rami Riziq Yousef Abumuaileq

---

## [Decision Letter · Decision Letter 1]

26 Apr 2024

A study protocol to characterise pathophysiological and molecular markers of rheumatic heart disease and degenerative aortic stenosis using multiparametric cardiovascular imaging and multiomics techniques

PONE-D-24-00937R1

Dear Dr. Mutithu,

We’re pleased to inform you that your manuscript has been judged scientifically suitable for publication and will be formally accepted for publication once it meets all outstanding technical requirements.

Kind regards,

Mohanad Alkhodari

Academic Editor

PLOS ONE

Additional Editor Comments (optional):

Reviewers' comments:

Reviewer's Responses to Questions

**Comments to the Author**

1. Does the manuscript provide a valid rationale for the proposed study, with clearly identified and justified research questions?

Reviewer #1: Yes

Reviewer #2: Yes

2. Is the protocol technically sound and planned in a manner that will lead to a meaningful outcome and allow testing the stated hypotheses?

Reviewer #1: Yes

Reviewer #2: Yes

3. Is the methodology feasible and described in sufficient detail to allow the work to be replicable?

Reviewer #1: Yes

Reviewer #2: Yes

4. Have the authors described where all data underlying the findings will be made available when the study is complete?

Reviewer #1: Yes

Reviewer #2: Yes

5. Is the manuscript presented in an intelligible fashion and written in standard English?

Reviewer #1: Yes

Reviewer #2: Yes

6. Review Comments to the Author

You may also provide optional suggestions and comments to authors that they might find helpful in planning their study.

Reviewer #1: The authors have modifed manuscript according to reviewers' suggestions. The manuscirpt is ready for publication.

Reviewer #2: The authors have addressed our comments and the manuscript has been improved. No further comments.

Regards

7. PLOS authors have the option to publish the peer review history of their article (what does this mean?). If published, this will include your full peer review and any attached files.

Reviewer #1: No

Reviewer #2: **Yes: **Rami Riziq Yousef Abumuaileq

---

## [Editor Report · Acceptance letter]

1 May 2024

PONE-D-24-00937R1 

PLOS ONE

Dear Dr. Mutithu, 

I'm pleased to inform you that your manuscript has been deemed suitable for publication in PLOS ONE. Congratulations! Your manuscript is now being handed over to our production team.

Kind regards, 

on behalf of

Dr. Mohanad Alkhodari 

Academic Editor

PLOS ONE